



# Revisiting chlorophyll extraction methods in biological soil crusts – methodology for determination of chlorophyll a and chlorophyll a+b as compared to previous methods

Jennifer Caesar[1,2,*], Alexandra Tamm[1,*], Nina Ruckteschler[1,*], Bettina Weber[1]

[1]Multiphase Chemistry Department, Max Planck Institute for Chemistry, Hahn-Meitner-Weg 1, D-55128 Mainz, Germany
[2]Present address: Department of Agriculture & Food Sciences, University of Applied Sciences, Brodaer Str. 2, D-17033 Neubrandenburg, Germany
[*]these authors contributed equally

*Correspondence to*: Jennifer Caesar (caesar@hs-nb.de), Bettina Weber (b.weber@mpic.de)

**Abstract.** Chlorophyll concentrations of biological soil crust (biocrust) samples are commonly determined to quantify the relevance of photosynthetically active organisms within these surface soil communities. Whereas chlorophyll extraction methods for freshwater algae and leaf tissues of vascular plants are well established, there is still some uncertainty regarding the optimal extraction method for biocrusts, where organism composition is highly variable and samples comprise major amounts of soil. In this study we analyzed the efficiency of two different chlorophyll extraction solvents, the effect of
grinding the soil samples prior to the extraction procedure and the impact of shaking as an intermediate step during extraction. The analyses were conducted on four different types of biocrusts. Our results show, that for all biocrust types chlorophyll contents obtained with ethanol were significantly lower than those obtained with dimethyl sulfoxide (DMSO) as solvent. Grinding of biocrust samples prior to analysis caused a highly significant decrease in chlorophyll content for green algal lichen- and cyanolichen-dominated biocrusts, and a tendency towards lower values for moss- and algae-dominated
biocrusts. Shaking of the samples after each extraction step had a significant positive effect on the chlorophyll content of green algal lichen- and cyanolichen-dominated biocrusts. Based on our results we confirm a DMSO-based chlorophyll extraction method without grinding pretreatment and suggest to insert an intermediate shaking step for complete chlorophyll extraction (see supplement S6 for detailed manual). Determination of a universal chlorophyll extraction method for biocrusts is essential for the inter-comparability of studies conducted across all continents.

## 1 Introduction

Chlorophyll (Chl) is a pigment commonly occurring in photosynthesizing organisms. It facilitates organisms to utilize sunlight as an energy source to build glucose and carbohydrates from $CO_2$ and water. The composition of photosynthetic pigments varies between organisms: whereas in cyanobacteria there is only $Chl_a$ and the accessory antenna pigments phycocyanin and phycoerythrin, green algae and vascular plants comprise $Chl_a$ and $Chl_b$, brown algae contain $Chl_c$ instead of
$Chl_b$ and rhodophyta contain only $Chl_a$ whereas $Chl_b$ is replaced by $Chl_d$.





Biological soil crusts (biocrusts) are surface soil communities commonly occurring in arid and semiarid regions throughout the world, as well as in areas where the lack of water or other environmental conditions (as e.g., disturbance) restrict the development of vascular plants (Garcia-Pichel et al., 2003; Zaady et al., 2016). They grow within the uppermost millimetres of the soil in close association with soil particles. It has been shown that biocrusts play significant functional roles in the

desert ecosystems (Eldridge and Greene, 1994; Evans and Belnap, 1999; Lan et al., 2011), as they stabilize the soil surface and reduce erosion by wind and water (Zhao et al., 2014; Belnap et al., 2014; Belnap and Büdel, 2016), they contribute to soil fertility through carbon and nitrogen fixation (Elbert et al., 2012; Sancho et al., 2016; Barger et al., 2016; Brankatschk et al., 2013), and they positively affect water retention and distribution in drylands (Rodriguez-Caballero et al., 2014; Chamizo et al., 2016). Biocrusts and their organisms have been shown to also release gaseous nitrogen compounds, as nitrous acid

(Lenhart et al., 2015), nitric oxide and nitrous oxide into the atmosphere (Weber et al., 2015; Meusel et al., 2017). Biocrusts are composed of photosynthesizing cyanobacteria, algae, lichens, and bryophytes plus decomposers, i.e. fungi, bacteria and archaea (Maier et al., 2016) and heterotrophic consumers, like protozoa, collembolans and snails (Darby and Neher, 2016; Bamforth, 2008). Thus, they form one of the smallest ecosystems with photosynthetic carbon fixation being the main source of carbohydrates. The chlorophyll content of these communities is therefore a good indicator for the photosynthetic capacity

and thus the capability of these systems to acquire energy, jointly used by the community but also exchanged with the surrounding environment. The capability to acquire energy under favorable environmental conditions is in turn a relevant proxy indicating successional stage, system stability, and its ability to recover from disturbance (Dojani et al., 2011; Gomez et al., 2012; Weber et al., 2016). As cyanobacteria and cyanolichens (photobiont: cyanobacteria) only comprise $Chl_a$, high $Chl_a/Chl_b$ ratios may indicate their dominance. Unfortunately, the $Chl_a/Chl_b$ ratio in algae, chlorolichens (photobiont: green

algae), and bryophytes (i.e. liverworts and mosses) is variable, thus not facilitating the proportional quantification of both groups; nevertheless, it allows tentative estimates on the relevance of eukaryotic $Chl_b$-comprising partners (Thorne et al., 1977).

Suitable chlorophyll extraction procedures are essential to obtain reliable results, and standardized methods are needed to allow comparison between studies (Schagerl and Künzl, 2007). During the last decades, several different methods have been

described. The solvents most commonly used for chlorophyll extraction have been ethanol, acetone, N, N-Dimethylformamide (DMF) and dimethyl sulfoxide (DMSO) (Mackinney, 1941; Shoaf and Lium, 1976; Moran and Porath, 1980; Barnes et al., 1992; Inskeep and Bloom, 1985). In a recent publication, Castle et al. (2011) compared the efficiency of four different solvents: acetone, ethanol, DMSO, and methanol for biological soil crusts of three different successional stages. They found that ethanol and DMSO extracted the greatest amount of $Chl_a$ using a double extraction technique. At a

similar time, Lan et al. (2011) also compared the $Chl_a$ extraction efficiency using ethanol, acetone, DMF and DMSO as solvents to analyse algal-, lichen-, and moss-dominated biocrusts. They concluded ethanol extractions expressed on an area basis to be most efficient and found DMSO extractions of lichens to be unreliable. Apart from the extraction solvent, also preparatory steps and handling during extraction varied between methods. Both Castle et al. (2011) and also Lan et al. (2011)





ground the samples with mortar and pestle to facilitate the following extraction steps, but Castle et al. (2011) also applied a two-step extraction and placed the samples on a shaker after each extraction cycle.

In this study we investigate these methodological techniques, which have been recently suggested for the extraction of $Chl_a$ from biocrusts. Our overall analytical technique is based on the photometric method established by Ronen and Galun (Ronen and Galun, 1984; Hiscox and Israelstam, 1978) and on methodological adaptions made in the lab of O.L. Lange (pers. comm.). We compare the efficiency of two extraction methods, using the two solvents DMSO and ethanol, which have been rated as most effective during the most recent studies by Lan et al. (2011) and Castle et al. (2011), for green algae-dominated, green algal lichen-dominated, moss-dominated, and cyanolichen-dominated biocrusts.

In this research we address the following three questions:

1) Which is the most potent chlorophyll extraction method for biocrusts? The ethanol method by Castle et al. (2011) or the DMSO method by Ronen and Galun (1984)?

2) Is a disruption of the cells (grinding) necessary prior to the extraction procedure and does it influence the chlorophyll yield?

3) Does shaking (20 minutes) after each extraction step influence the chlorophyll yield?

The overall goal of this study is to determine an extraction method best suited for chlorophyll determination of the analyzed samples, which we consider as an important step towards a universal chlorophyll determination method for different types of biocrusts.

## 2. Material and Methods

### 2.1 Sampling site

The biocrust samples for this study were collected in the nature conservation area "Mehlinger Heide", located in Rhineland-Palatinate, about 15 kilometres north of the city Kaiserslautern, and in the nature reserve "Ruine Homburg" at Gössenheim, in northern Bavaria.

The "Mehlinger Heide" is about 410 ha in size, being one of the biggest heathlands in southern Germany. Until 1912, the area was completely covered by forest. In the First and Second World War, the region was partly deforested to build a military training ground (http://mehlinger-heide.de/). After the Second World War, the area was used for military training by French and US American troops until 1992 and 1994, respectively. The prolonged continuous disturbance by military use caused the formation of a special flora and fauna typical for nutrient-depleted sandy soils, which is only rarely found in central Europe. The vegetation is characterized by dwarf shrubs (heather), lichens (fruticose, foliose and crustose) and bryophytes (mosses and liverworts). Since 2001, the "Mehlinger Heide" is under conservation.

The nature reserve "Ruine Homburg" near Gössenheim, Germany, is an open anthropogenic landscape with bare rock and gravel spots covered by a thin vegetation layer dominated by cryptogams including lichens, bryophytes, and cyanobacteria. Its bedrock is Triassic shell limestone and its flora is composed of a relic flora after the ice age with sub-mediterranean-



continental and sub-mediterranean-sub-atlantic elements (Lösch, 1980). The landscape has remained open because of a castle which was built nearby in 1080.

## 2.2 Sample collection

Four different biocrust types were collected:

1) green algae-dominated biocrust with *Klebsormidium* spec. as dominating organism

2) green algal lichen-dominated biocrust with *Cladonia* spec. as dominating genus

3) moss-dominated biocrust with *Hypnum* spec. as dominating organism

4) cyanolichen-dominated biocrust with *Peltigera rufescens* as dominating lichen

All four biocrusts types were collected in 2014 (January and May) and 2016 (June and September) in the "Mehlinger Heide". Some samples of the cyanolichen-dominated biocrusts were collected in May 2014 in Aschfeld/Gössenheim. Within each experiment, only samples of the same sampling batch were used. During sampling, special care was taken to avoid variability between replicate samples within each type. For sampling, a metal ring of 14 mm diameter (surface size: 153.9 mm²) and 3.5 cm in height was used. The metal ring was pressed 3 cm deep into the soil, a trowel was pushed underneath,

both were pulled out again and the biocrust sample within the ring was transferred into a plastic zip lock bag. For each biocrust type 5 replicates were collected (total amount of samples: n = 20). The samples were transported back to the institute, where they were air dried and stored at room temperature at low light intensities for less than 4 weeks until the chlorophyll extraction experiments were conducted.

## 2.3 Chlorophyll extraction

### 2.3.1 Pretreatment of the biocrust samples

Prior to chlorophyll extraction, all biocrust samples were dried in a drying oven at 60°C for at least 24 hours until constant weight was reached and the dry weight was determined. On the day before the chlorophyll extraction the samples were slightly sprinkled with distilled water to activate the biocrust organisms, which is known to facilitate the subsequent chlorophyll extraction. The photosynthetic apparatus undergoes disassembly during desiccation, so rehydration is required to

repair the photosynthetic apparatus (Harel et al., 2004). During the entire chlorophyll extraction procedure, the samples were kept in the dark or at minimum light to prevent degradation of chlorophyll (Molnár et al., 2013; Hosikian et al., 2010).

### 2.3.2 Chlorophyll extraction with dimethyl sulfoxide (DMSO)

Chlorophyll extraction with DMSO as solvent was conducted based on the method described by Ronen and Galun (Hiscox and Israelstam, 1978; Ronen and Galun, 1984) and on methodological adaptions made in the lab of O.L. Lange (pers.

comm.).



The soil crust samples of the four types (n = 5) were placed in 15 mL screw-cap vials without prior grinding. A spatula tip of $MgCO_3$ or $CaCO_3$ was added to avoid acidification and the associated chlorophyll degradation (Weber et al., 2013; Rapsch and Ascaso, 1985). Then 6 mL DMSO (ROTIDRY® ≥ 99.5 %, ≤ 200 ppm $H_2O$, Carl Roth GmbH + Co. KG, Karlsruhe, Germany) were added to the samples. A vial without sample but DMSO and $MgCO_3$/$CaCO_3$ was used as blank. All vials

were placed in a water bath at 65°C for 90 minutes. The caps of the vials were half tightened to allow extension of the liquid during heating but avoid evaporation loss. After the first extraction cycle the supernatant was poured into a separate vial, and another 6 mL of DMSO was added to the samples for a second extraction cycle. After a second extraction in the water bath for 90 minutes, the supernatants of both extractions were pooled and centrifuged (Technospin R, Sorvall Instruments, Darmstadt, Germany) for 5 minutes at 4000 rpm and 15°C before photometric determination.

The absorption was measured with a spectrophotometer (Lambda 25 UV/VIS, PerkinElmer, Rodgau, Germany) at 648, 665 and 700 nm. If absorption values at 665 nm were above 0.8, the sample was diluted 1:1 with DMSO and the equation was adjusted accordingly, changing the dilution factor from 1 to 2. The $Chl_{a+b}$ concentrations were calculated according to O.L. Lange (pers. comm.).

Total $Chl_{a+b}$ amount in sample:

$Chl_{a+b}$ [µg] = [($A_{665}$ − $A_{700}$) * 8.02 + ($A_{648}$ − $A_{700}$) * 20.2] * DF * S      (1)

$Chl_{a+b}$ amount based on surface area:

$Chl_{a+b}$ [mg * $m^{-2}$] = $Chl_{a+b}$ [µg] / AR / 1000      (2)

$Chl_{a+b}$ amount based on dry weight:

$Chl_{a+b}$ [µg * $g^{-1}$] = $Chl_{a+b}$ [µg] / DW      (3)

where

$A_\lambda$ =    absorbance at certain wavelength,

DF =    dilution factor,

S  =    amount of solvent [mL],

AR =    area [$m^2$], the area in this case is always 153.9 mm², as the radius of the sampling ring is 7 mm (AR = $\pi$ * $r^2$),

DW =    dry weight [g].

The $Chl_a$ concentration of biocrusts dominated by cyanobacterial lichens was calculated according to Arnon et al. (1949).

Total $Chl_a$ amount in sample:

$Chl_a$ [µg] = [($A_{665}$ − $A_{700}$) * 12.19] * DF * S      (4)

$Chl_a$ amount based on surface area:

$Chl_a$ [mg * $m^{-2}$] = $Chl_a$ [µg] / AR / 1000      (5)

$Chl_a$ amount based on dry weight:

$Chl_a$ [µg * $g^{-1}$] = $Chl_a$ [µg] / DW      (6)



### 2.3.3 Reproducibility of chlorophyll extraction with dimethyl sulfoxide (DMSO)

In order to analyze the reproducibility of chlorophyll extraction with DMSO, ~ 90 g of green algae-dominated biocrust were oven-dried (see 2.3.1), homogenized with a mortar and pestle and evenly distributed into eight screw-cap vials. The subsequent chlorophyll extraction followed the described DMSO extraction procedure (see 2.3.2). The analyzed chlorophyll

content refers to the initial sample weight.

### 2.3.4 Chlorophyll extraction with ethanol

The $Chl_a$ double extraction with ethanol was carried out according to the protocol of Castle et al. (2011). The soil crust samples (four types, n = 5) were ground with mortar and pestle until they were homogenous and placed in screw-cap vials, a spatula tip of MgCO3 and 6 mL ethanol (≥ 99.8 %, Sigma-Aldrich, Steinheim, Germany) were added and the caps were half-

tightened. A vial with ethanol and a spatula tip of MgCO3 was used as blank. All samples were heated in a water bath at 80°C until they started to boil. Once they had started to boil they were kept in the water bath for 5 minutes. After boiling, the vials were cooled down for 10 minutes and then placed in a horizontal shaker for 20 minutes. Subsequently, the samples were centrifuged at 4000 rpm and 15°C for 10 minutes. The supernatant was poured into separate vials and another 6 mL ethanol were added to the samples for a second extraction cycle conducted in the same way as the first one.

After the second cycle, the supernatants of both extractions were combined and the absorbance was measured spectrophotometrically at 665, 649 and 750 nm. The absorption at 750 nm was measured in addition to eliminate a potential effect of the intrinsic color of the samples. The $Chl_{a+b}$ content was determined according to the formula of Ritchie (2006).

Total Chl amount in sample:

$$Chl_a \, [\mu g] = 13.5275 * (A665 - A750) - 5.201 * (A649 - A750) * DF * S \tag{7}$$

$$Chl_b \, [\mu g] = 22.4327 * (A649 - A750) - 7.0741 * (A665 - A750) * DF * S \tag{8}$$

The total $Chl_{a+b}$ content is the sum of the results obtained from formula (7) and (8).

Analogous to the DMSO method, the total chlorophyll content can be calculated per area by dividing the total amount of $Chl_{a+b}$ by the area (AR) (see also 2.3.2).

Chl amount based on surface area:

$$Chl_{a+b} \, [mg * m^{-2}] = Chl_{a+b} / AR / 1000 \tag{9}$$

The $Chl_a$ concentration of biocrusts dominated by cyanobacterial lichens was calculated according to the formula by Ritchie (2006).

Total $Chl_a$ amount in sample:

$$Chl_a \, [\mu g] = [11.9035 * (A665 - A750)] * DF * S \tag{10}$$

$Chl_a$ amount based on surface area:

$$Chl_a \, [mg * m^{-2}] = Chl_a \, [\mu g] / AR / 1000 \tag{11}$$



### 2.4 Effect of prior grinding and additional shaking on chlorophyll extraction efficiency

Additional experiments were performed to evaluate the effect of two methodological steps, i.e. grinding prior to extraction and shaking after each extraction cycle of the samples, as described in the protocol by Castle et al. (2011). These experiments were only carried out with the DMSO method described above. To analyze the effect of grinding, the samples

were ground to homogeneity with a mortar and pestle prior to the extraction procedure. In a second approach, the effect of shaking of the samples after extraction was analyzed. For that, the samples were placed on a horizontal shaker for 20 minutes after each extraction cycle. Both treatments were applied to the four biocrust types (n = 5; in shaker experiment: green algal- and green algal lichen-dominated biocrust: n = 4).

### 2.5 Statistical evaluation

All data were analyzed with the statistical and analytical software OriginPro (Version 8.6; OriginLab Corporation, Northampton, MA, USA). Before statistical analyses all data were tested for normality and variance homogeneity. To determine statistical differences between normally distributed samples, where homogeneity of variance was given, paired t-tests were performed. For samples, where normal distribution could not be reached, a Mann-Whitney U-test was performed. Normally distributed samples without homogeneity of variance were analyzed using the Welch test.

For some samples the relative standard deviation (RSD) was calculated to reveal the extent of variability in relation to the average value of the population, using the following formula:

$$RSD = \sigma / \mu * 100 \% \tag{12}$$

where is $\sigma$ = standard deviation and $\mu$ = average value.

The RSD indicates the precision of the data and shows if the data is tightly clustered around the mean.

## 3. Results

### 3.1 Comparison of chlorophyll extraction methods: DMSO versus ethanol

The two different chlorophyll extraction methods for DMSO and ethanol showed significant differences in chlorophyll yield for all four biocrust types (Fig. 1). Using the DMSO method, significantly higher chlorophyll contents were obtained compared to the ethanol method. Whereas $Chl_{a+b}$ contents determined with the DMSO extraction method ranged between

mean values of ~ 550 and 900 mg m$^{-2}$ (i.e. 598 mg m$^{-2}$ for green algae-dominated, 547 mg m$^{-2}$ for cyanolichen-dominated, 772 mg m$^{-2}$ for green algal lichen-dominated and 903 mg m$^{-2}$ for moss-dominated biocrusts), values obtained by the ethanol extraction method were between ~ 110 and 370 mg m$^{-2}$ (i.e., 182 mg m$^{-2}$ for green algae-dominated, 111 mg m$^{-2}$ for cyanolichen-dominated, 125 mg m$^{-2}$ for green algal lichen-dominated and 367 mg m$^{-2}$ for moss-dominated biocrusts; Supplement Table S1). Thus, the mean values obtained by the ethanol extraction procedure were ~ 70, 80, 84 and 59 %

lower for green algae-, cyanolichen-, green algal lichen- and moss-dominated biocrusts than those obtained by the DMSO



method. However, for all biocrust types, the results showed higher standard deviations for the DMSO as compared to the ethanol extraction method, and all types had higher relative standard deviations (Supplement Table S2).

To evaluate reproducibility of the chlorophyll extraction procedure by means of the DMSO method, eight replicate samples of homogeneous biocrust material were analyzed. Here, a RSD of 12.1 % was obtained, certifying a good reproducibility of

5   the DMSO extraction method (Supplement Table S3).

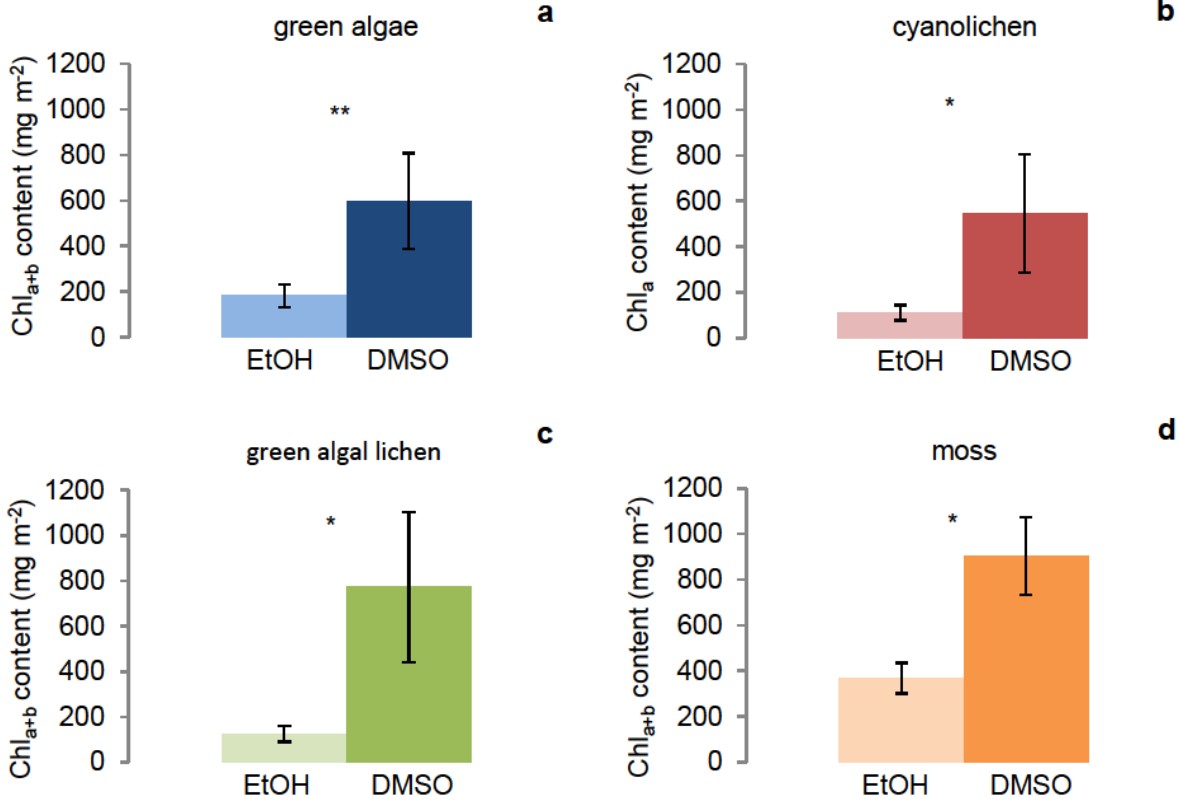

**Figure 1: Chlorophyll$_{a+b}$ content (mg m$^{-2}$) of green algae- (a), cyanolichen- (b), green algal lichen- (c) and moss-dominated biocrusts (d) using the DMSO and ethanol extraction method. To prove statistical differences between both extraction methods,**

10   **the Mann-Whitney U-test was performed for moss-dominated biocrusts, as normal distribution of the data was not given. For the other three biocrust types the Welch test was performed, as the homogeneity of variance was not given for the data (\*: p ≤ 0.05; \*\*: p ≤ 0.01; \*\*\*: p ≤ 0.001).**

### 3.2 Pretreatment of the samples: Grinding versus non-grinding

As the ethanol extraction method contains an additional grinding step during sample pretreatment, the impact of this step

15   was tested for the DMSO extraction method, which had gained higher chlorophyll extraction yields (Fig. 1). The results reveal that preparatory grinding caused a decrease in Chl$_{a+b}$ contents, causing significantly lower values for green algal lichen- and cyanolichen- and a similar tendency in green algae- and moss-dominated biocrusts (Fig. 2). In fact, this





preparatory step caused mean Chl$_{a+b}$ yields to be ~ 26, 36, 51 and 25 % lower in green algae-, cyanolichen-, green algal lichen- and moss-dominated biocrusts (Supplement Table S4).

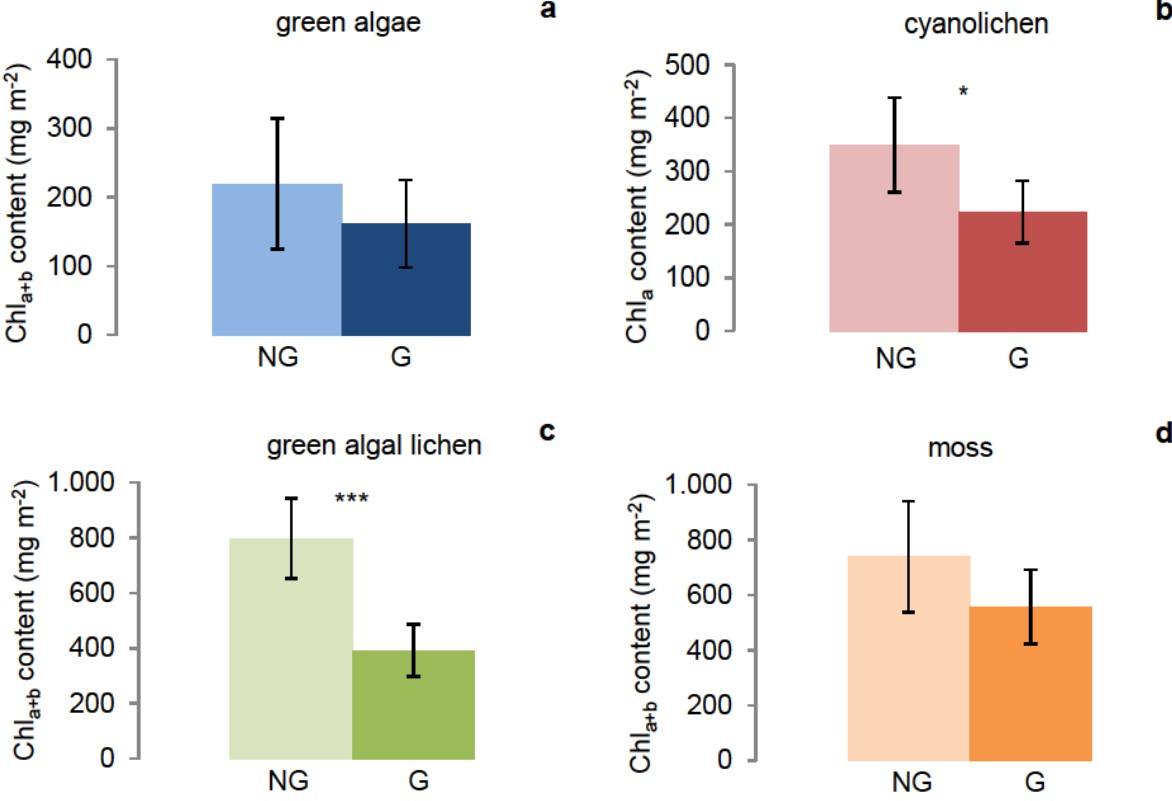

5  **Figure 2: Chlorophyll$_{a+b}$ content (mg m$^{-2}$) depending on preparatory grinding (G) as compared to control extractions without grinding (NG). Investigation of green algae- (a), cyanolichen- (b), green algal lichen- (c) and moss-dominated biocrusts (d) using the DMSO extraction method. To prove statistical differences, the paired t-test was performed for all four biocrust types. (*: p ≤ 0.05; **: p ≤ 0.01; ***: p ≤ 0.001).**

**3.3 Intermediate step during chlorophyll extraction: Shaking versus non-shaking**

10  The effect of shaking after each extraction cycle was evaluated for all four crust types. As shown in Fig. 3, shaking caused increased chlorophyll yields in green algal lichen- and cyanolichen-dominated biocrusts and a trend in the same direction was observed for green algae-dominated crusts. In moss-dominated biocrusts shaking had no effect. Chl$_{a+b}$ values were ~ 39, 73, and 42 % higher in green algae-, cyanolichen-, and green algal lichen-dominated biocrusts (Supplement Table S5).





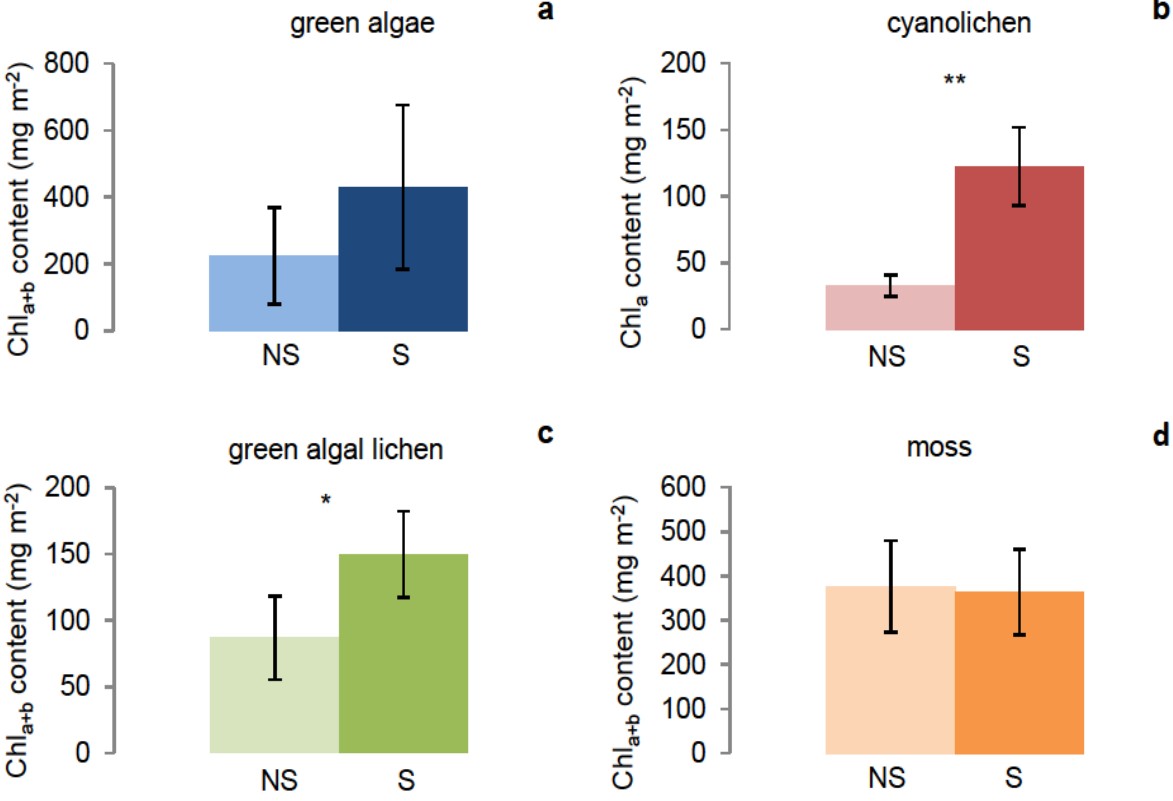

**Figure 3: Chlorophyll$_{a+b}$ content (mg m$^{-2}$) depending on intermediate shaking (S) after each extraction cycle as compared to control extractions without shaking (NS). Investigation of green algae- (a), cyanolichen- (b), green algal lichen- (c) and moss-dominated biocrusts (d) using the DMSO extraction method. To prove statistical differences between shaking (S) and not shaking (NS) of the samples, for cyanolichen- and moss-dominated biocrusts the paired t-test and for green algae- and green algal lichen-dominated biocrusts the unpaired t-test were performed. (*: p ≤ 0.05; **: p ≤ 0.01; ***: p ≤ 0.001).**

## 4. Discussion

Whereas biocrusts represent an important component of the landscape in arid and semi-arid environments and are a natural and most effective force in land stabilization and recovery (Campbell, 1979; Belnap et al., 2003; Weber et al., 2016), no universal method has been established, yet, to determine the chlorophyll content of the photosynthesizers within these microbial communities. In the present study, we evaluated the usefulness of different preparatory steps and methods proposed in recent studies to then determine a chlorophyll extraction technique most suitable for biocrust samples. A perfect extraction procedure should deliver rapid and reproducible results, the used solvent should bring all pigments into solution and has to be simple to execute, resolve pigments to extremely low levels of detection, be hazard-free, and cause no chemical changes to the pigments (Jeffrey, 1981).





Our investigations illustrate, that DMSO extracted $Chl_a$ and $Chl_{a+b}$ pigments to a significantly larger extent than ethanol. Differences for cyanolichen-, green algal lichen- and moss-dominated biocrusts were significant, those for green algae-dominated biocrusts even highly significant. Grinding of samples prior to the extraction procedure had a significant negative effect on the extracted amounts of chlorophyll, whereas shaking of samples after each extraction cycle caused significantly

increased chlorophyll contents for most biocrust types.

Already Hiscox and Israelstam (1978) described that DMSO is applicable for a wide range of plant types with variable leaf tissues, and also Lan et al. (2011) measured the highest extraction efficiency for DMSO when analyzing biocrusts. Using the DMSO method, the duration of the chlorophyll extraction procedure is fairy uncritical, as after extraction the samples can even be stored in the fridge overnight and spectrometrically analyzed on the next day (Barnes et al., 1992). During

chlorophyll extraction with DMSO, $CaCO_3$ is added to prevent acidification and minimize the phaeophytinization of $Chl_{a+b}$ (Barnes et al., 1992), which otherwise happens easily, as chlorophyll is sensitive to extreme light exposure, pH values and temperature (Molnár et al., 2013; Hosikian et al., 2010). In studies, where acidification has not been prevented, unnaturally high absorption values were observed for lichen samples around 665 nm (Lan et al., 2011), where also phaeophytin is absorbing (Ritchie, 2008), thus suggesting phaeophytinisation of $Chl_a$ by lichen acids. Solvents containing methanol, ethanol

or 1-propanol are also known to easily degrade $Chl_a$ and $Chl_{a+b}$, as isomerisation and allomerization of chlorophyll molecules occurs very easily under acidic conditions (Hynninen, 1977), which need to be avoided by the addition of magnesium carbonate (Ritchie, 2008). The extraction efficiency of DMSO was higher, but also showed larger variability between replicates compared to the ethanol extraction method (Supplement Table S1). As thorough chlorophyll extraction has been reported for DMSO as solvent in previous studies (Barnes et al., 1992; Castle et al., 2011) and a good reproducibility of the

results has been shown for biocrusts by us (Supplement Table S3) and for leaf fragments by Tait and Hik (2003), we believe that this variability reflects the actual variability in chlorophyll contents between samples. In contrast to the extraction efficiency, ethanol has the advantage of being non-toxic (Lan et al., 2011), whereas DMSO has a potential to carry dissolved substances into the body through the skin (Horita and Weber, 1964; Sulzberger et al., 1967).

Grinding the biocrust samples before extraction had a particularly negative effect on green algal lichen- and cyanolichen-

dominated biocrusts, which seem to be particularly sensitive towards the damage of cells leading to chlorophyll degradation. This again could be caused by lichen acids. The biocrust samples in the current experiments were ground by hand with a mortar and pestle and no high temperatures were reached. In other experiments, cooling with ice may be necessary to avoid overheating, as chlorophyll is easily degraded by heat (Braumann and Grimme, 1979). If the cells are being disrupted within less than 20 seconds very rapidly and efficiently by a motor driven pestle, cooling with ice may also not be necessary

(Schagerl and Künzl, 2007). Loss of extraction efficiency due to increasing interference of humus, polysaccharides and clay also cannot be excluded.

Shaking of biocrust samples after each extraction cycle had a mostly positive effect on extraction efficiency, as mean extraction quantities were significantly higher for cyanolichen- and chlorolichen-dominated biocrusts, and showed the same tendency for green algal-dominated biocrusts. Solely for moss-dominated biocrusts shaking of samples had no effect. An



effectiveness of shaking has also been shown for other substances, as the extraction of polyphenols from basil leaves and of toxic elements from artificial saliva (Zlotek et al., 2016; Arain et al., 2013).

Error sources, which may cause an increased variability of the chlorophyll contents, are differences in sample size and composition. Whereas variation in sample size seems negligible, as defined sampling rings of fixed area and height were used, variation in sample growth could not be completely excluded, although during sampling special care was taken to minimize this effect.

## 5. Conclusion

In conclusion, the determination of chlorophyll content using DMSO as solvent ensures a simple, rapid and stable extraction. A cell-disrupting pretreatment, like grinding or homogenization of the samples, is not required for a complete chlorophyll extraction and can even cause chlorophyll loss as seen for the lichen-dominated biocrusts. Moreover, this turned out to be a time consuming step, especially when large sample numbers are processed. In contrast, shaking between two extraction cycles turned out to be positive. An advantage of chlorophyll extraction with DMSO is that the samples are stable over 6 to 10 days after incubation and can be stored at 4 to 8°C without degradation of the pigments (Ronen and Galun, 1984; Barnes et al., 1992), while in other solvents significant amounts of chlorophyll are lost during storage (Hiscox and Israelstam, 1978). Thus, based on our experiments, we developed a universal DMSO-based chlorophyll extraction method for biocrusts (Supplement S6).

*Data availability.* Mean values and standard deviations of all data are listed in the supplement. Raw data can be obtained from the corresponding authors upon request.

*Author contribution.* BW, JC, AT, and NR designed the experiments. AT and NR collected soil samples and carried out the laboratory work. JC and BW prepared the manuscript with contributions from all co-authors.

*Competing interests.* The authors declare that they have no conflict of interest.

*Acknowledgements.* This work was supported by the Max Planck Society. BW would like to thank Paul Crutzen for the award of a Nobel Laureate Fellowship (2013 – 2015), and was financed by the German Research Foundation (DFG-FOR 1525: INUIT; WE2393/2). We would like to thank Anna Lena Leifke, Jens Weber and Heike Pfaff for their help during lab work. For field work at the study sites, permissions were obtained from the Untere Naturschutzbehörde Kaiserslautern and the Regierung Unterfranken (in the framework of the SCIN project).



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
