# Peer review of "Revisiting chlorophyll extraction methods in biological soil crusts – methodology for determination of chlorophyll a and chlorophyll a+b as compared to previous methods"

_Biogeosciences, 2017_

## Referee Comment (RC1) · J. Belnap (Referee) · 18 Oct 2017

1.  Some place have awkward English (e.g., "To prove statistical differences between both extraction methods, the Mann-Whitney U-test was performed for moss-dominated biocrusts, as normal distribution of the data was not given." I believe what is meant is that the distribution was not normal" and "Solely for moss-dominated biocrusts shaking of samples had no effect.")

2.  One cannot say values are higher or lower if they are not statistically different. In

the Results: "The results reveal that preparatory grinding caused a decrease in Chla+b contents, causing significantly lower values for green algal lichen- and cyanolichen- and a similar tendency in green algae- and moss-dominated biocrusts (Fig. 2). In fact, this preparatory step caused mean Chla+b yields to be ∼ 26, 36, 51 and 25 % lower in green algae-, cyanolichen-, green algal lichen- and moss-dominated biocrusts (Supplement Table S4)." If the values for green algae- and moss-dominated biocrusts are not statistically distinct among treatments, one cannot say "…this step caused mean Chla+b yields to be ∼ 26, 36, 51 and 25 % lower in green algae-, cyanolichen-, green algal lichen- and moss-dominated biocrusts (Supplement Table S4)". Same for this sentence "In moss-dominated biocrusts shaking had no effect. Chla+b values were ∼ 39, 73, and 42 % higher in green algae-, cyanolichen-, and green algal lichen-dominated biocrusts (Supplement Table S5).

3. "Shaking of biocrust samples after each extraction cycle had a mostly positive effect on extraction efficiency, as mean extraction quantities were significantly higher for cyanolichen- and chlorolichen-dominated biocrusts, and showed the same tendency for green algal-dominated biocrusts: As only 2/4 were enhanced, I would say "affected extraction efficiency", not "mostly positive".

4. Discussion and Conclusions: Several other labs have also tested different extraction techniques and have reached a different conclusion regarding the best extractant and whether to grind samples. These need to be acknowledged and discussed more thoroughly in both the Discussion and Conclusions, as there is still work to be done to clarify why different results are being obtained. This especially applies to grinding, as I know of at least 4 labs testing grinding versus not, and they all obtained higher chlorophyll extraction with grinding. My lab has tested this multiple times, and even included hand vs mill, cold vs not cold, and still obtained higher values with any kind of grinding (although hand grinding was superior to the mill). The difference may be that we are using a cyanobacterially-dominated biocrust that contains 0-15% green algal and cyanolichens, but I cannot think of why that would affect this question. Regardless,

there is clearly further work to be done and I would thus not end with "Thus, based on our experiments, we developed a universal DMSO-based chlorophyll extraction method for biocrusts." Instead, this needs to be state that this issue is still not resolved.

---

## Referee Comment (RC2) · K. J. van Groenigen (Referee) · 29 Nov 2017

Caesar et al. present the results of a study in which they assessed several methodologies to extract chlorophyll from biocrusts. In a well-organized experiment, they compared the effect of variations on the key steps throughout the extraction process: the solvent that was used (ethanol vs. dmso), grinding vs. no grinding, and shaking vs. no shaking. The results are presented clearly, and the discussion is on point. I second the comments and suggestions by reviewer 1, and I only have a few additional minor comments and suggestions, all of which are easy to address:

[Figure]

page 1, l30: please provide a reference to support this statement. page 2, l2: delete "as", i.e. "...conditions (e.g. disturbance).." page 2, l4: delete "the" from "the desert ecosystems" page 10, l12-16:this sentence is grammatically incorrect. Please consider the following alternative:"A perfect extraction procedure should deliver rapid and reproducible results and has to be simple to execute. Furthermore, the used solvent should bring all pigments into solution, resolve pigments to extremely low levels of detection, be hazard-free, and cause no chemical changes to the pigments (Jeffrey, 1981)." page 11, l5: is "most types" really appropriate here? It seems like this is only true for 2 out of 4 biocrusts. page 11, l6: this sentence is grammatically incorrect. Please consider the following alternative: "Hisco page 11, l8: "fairy uncritical, as after.." should be "fairly uncritical; after.. ", or even better: "less critical; after.." page 11, l31: please provide a reference to support this statement page 12, l12: what do you mean by "turned out to be positive"? Improved extraction efficiency?

---

## Author Comment (AC1) · 27 Dec 2017

First of all we would like to thank you for your helpful comments and suggestions to improve our manuscript. We have reviewed all your comments and revised the manuscript as seen below:

1. Some place have awkward English (e.g., "To prove statistical differences between both extraction methods, the Mann-Whitney U-test was performed for moss-dominated biocrusts, as normal distribution of the data was not given." I believe what is meant is

that the distribution was not normal" and "Solely for moss-dominated biocrusts shaking of samples had no effect.")

We corrected the caption of Figure 1 (page8, line 9-11) as written below and also checked the writing of the manuscript once again. "To prove statistical differences between both extraction methods, the Mann-Whitney U-test was performed for moss-dominated biocrusts (data not normally distributed), for the other three biocrust types the Welch test was performed (data without homogeneity of variance) (*: $p \leq 0.05$; **: $p \leq 0.01$; ***: $p \leq 0.001$)."

2. One cannot say values are higher or lower if they are not statistically different. In the Results: "The results reveal that preparatory grinding caused a decrease in Chla+b contents, causing significantly lower values for green algal lichen- and cyanolichen- and a similar tendency in green algae- and moss-dominated biocrusts (Fig. 2). In fact, this preparatory step caused mean Chla+b yields to be âĹij 26, 36, 51 and 25 % lower in green algae-, cyanolichen-, green algal lichen- and moss-dominated biocrusts (Supplement Table S4)." If the values for green algae- and moss-dominated biocrusts are not statistically distinct among treatments, one cannot say "...this step caused mean Chla+b yields to be âĹij26, 36, 51 and 25 % lower in green algae-, cyanolichen-, green algal lichen- and moss-dominated biocrusts (Supplement Table S4)". Same for this sentence "In moss-dominated biocrusts shaking had no effect. Chla+b values were âĹij39, 73, and 42 % higher in green algae-, cyanolichen-, and green algal lichen-dominated biocrusts (Supplement Table S5).

Thank you very much for this correct comment. We altered these sentences and only point out the percentages for the significant changes.

3. "Shaking of biocrust samples after each extraction cycle had a mostly positive effect on extraction efficiency, as mean extraction quantities were significantly higher for cyanolichen- and chlorolichen-dominated biocrusts, and showed the same tendency for green algal-dominated biocrusts: As only 2/4 were enhanced, I would say "affected

extraction efficiency", not "mostly positive".

Thank you for this suggestion. We corrected the sentence accordingly (page 11, line 33).

4. Discussion and Conclusions: Several other labs have also tested different extraction techniques and have reached a different conclusion regarding the best extractant and whether to grind samples. These need to be acknowledged and discussed more thoroughly in both the Discussion and Conclusions, as there is still work to be done to clarify why different results are being obtained. This especially applies to grinding, as I know of at least 4 labs testing grinding versus not, and they all obtained higher chlorophyll extraction with grinding. My lab has tested this multiple times, and even included hand vs mill, cold vs not cold, and still obtained higher values with any kind of grinding (although hand grinding was superior to the mill). The difference may be that we are using a cyanobacterially-dominated biocrust that contains 0-15% green algal and cyanolichens, but I cannot think of why that would affect this question. Regardless, there is clearly further work to be done and I would thus not end with "Thus, based on our experiments, we developed a universal DMSO-based chlorophyll extraction method for biocrusts." Instead, this needs to be state that this issue is still not resolved.

After we read this comment, we reconsidered our grinding method and wondered if the results were different if we ground our samples in a wet stage with extraction solvent being present. Thus, we ran an additional experiment comparing samples hand-ground in a wet stage in a mortar to non-ground samples, but we again obtained the result that grinding had a negative effect on the chlorophyll content. We included this additional experiment in the manuscript. As these different results obtained by different labs indeed demand for further research on this topic, we added this information in the discussion and altered our final sentence of the conclusion in the following way: Discussion, page 11, line 32 ff.: "...In contrast to our results, in other labs grinding was observed to improve chlorophyll extraction efficiency (J. Belnap, pers. comm.)..." Conclusion, page 12, line 18 ff.: "...Thus, based on our experiments, we developed

a DMSO-based chlorophyll extraction method optimized for green algae-, lichen-, and moss-dominated biocrusts (Supplement S6). . ."
* * *

---

## Author Comment (AC2) · 27 Dec 2017

First of all we would like to thank you for your evaluation of our manuscript Thanks for the helpful comments and suggestions to improve our manuscript.

We have reviewed all your comments and suggestions and revised the manuscript as seen below:

Caesar et al. present the results of a study in which they assessed several methodologies to extract chlorophyll from biocrusts. In a well-organized experiment, they compared the effect of variations on the key steps throughout the extraction process: the solvent that was used (ethanol vs. dmso), grinding vs. no grinding, and shaking vs. no shaking. The results are presented clearly, and the discussion is on point. I second the comments and suggestions by reviewer 1, and I only have a few additional minor comments and suggestions, all of which are easy to address:

-We would like to thank you for this very positive evaluation of our manuscript.

1. Please provide a reference to support this statement (page 1, l30).

- We now cite the following textbook: Mohr, H., Schopfer, P. (1995) Plant Physiology. 4th edition. Springer-Verlag Berlin, Heidelberg, New York.

2. Delete "as", i.e. "...conditions (e.g. disturbance).." (page 2, l4).

- Thank you for mentioning. We did the correction as suggested.

3. page 2, l4: delete "the" from "the desert ecosystems"

- Done as suggested.

4. This sentence is grammatically incorrect (page 10, l12-16). Please consider the following alternative: "A perfect extraction procedure should deliver rapid and reproducible results and has to be simple to execute. Furthermore, the used solvent should bring all pigments into solution, resolve pigments to extremely low levels of detection, be hazard-free, and cause no chemical changes to the pigments (Jeffrey, 1981)".

-Thanks for offering this alternative. The sentence has been corrected accordingly.

5. Is "most types" really appropriate here? It seems like this is only true for 2 out of 4 biocrusts (page 11, l5).

- You are right. We altered the sentence in the following way: "...Grinding of samples prior to the extraction procedure had a significant negative effect on the extracted amounts of chlorophyll, whereas shaking of samples after each extraction cycle caused

significantly increased chlorophyll contents for two of four biocrust types. . ."

6. This sentence is grammatically incorrect. Please consider the following alternative: "Fairy uncritical, as after.." should be "fairly uncritical; after.. ", or even better: "less critical; after.." (page 11, l8).

- We changed the sentence according to your last proposal "less critical; after..."

7. Please provide a reference to support this statement (page 11, l31).

- We inserted the reference "Lan, S. B., Wu, L., Zhang, D. L., Hu, C. X., and Liu, Y. D.: Ethanol outperforms multiple solvents in the extraction of chlorophyll-a from biological soil crusts, Soil Biology & Biochemistry, 43, 857-861, doi:10.1016/j.soilbio.2010.12.007, 2011" to support the statement as suggested (page 11, line 32).

8. What do you mean by "turned out to be positive" (page 12, l12)? Improved extraction efficiency?

- Yes, you are right. That is what we meant and thus we changed the sentence in the following way: ". . .In contrast, shaking between two extraction cycles turned out to improve extraction efficiency. . ."